

# Quantitative trait loci associated with soft wheat quality in a cross of good by moderate quality parents

Gioconda Garcia-Santamaria[1,*], Duc Hua[2] and Clay Sneller[2,*]

[1] Faculty of Agricultural Sciences, Central University of Ecuador, Quito, Pichincha, Ecuador
[2] Department of Horticulture and Crop Sciences, Ohio State University—Wooster, Wooster, OH, United States of America
[*] These authors contributed equally to this work.

## ABSTRACT

Information on the genetic control of the quality traits of soft wheat (*Triticum aestivum*) is essential for breeding. Our objective was to identify QTL associated with end-use quality. We developed 150 F4-derived lines from a cross of Pioneer 26R46 × SS550 and tested them in four environments. We measured flour yield (FY), softness equivalent (SE), test weight (TW), flour protein content (FP), alkaline water retention capacity (AWRC), and solvent retention capacity (SRC) of water (WA), lactic acid (LA), sucrose (SU), sodium carbonate (SO). Parents differed for nine traits, transgressive segregants were noted, and heritability was high (0.67 to 0.90) for all traits. We detected QTL distributed on eight genomic regions. The QTL with the greatest effects were located on chromosome 1A, 1B, and 6B with each affecting at least five of ten quality traits. Pioneer 26R46 is one of the best quality soft wheats. The large-effect QTL on 1A novel and accounted for much of the variation for AWRC ($r^2 = 0.26$), SO (0.26) and SE (0.25), and FY (0.15) and may explain why Pioneer 26R46 has such superior quality. All alleles that increased a trait came from the parent with the highest trait value. This suggests that in any population that marker-assisted selection for these quality traits could be conducted by simply selecting for the alleles at key loci from the parent with the best phenotype without prior mapping.

## INTRODUCTION

Soft red winter wheat (SRWW) (*Triticum aestivum* L. em. Thell) end-use quality is determined by flour quality requirements related to grain characteristics and flour functionality. Functional flour for US biscuit industry should have a low water absorption capacity, high gluten strength, low damaged starch and arabinoxylans whereas for bread making needs high water absorption capacity, good gluten strength and high damaged starch and arabinoxylans or the so called water extractable arabinoxylans (*Slade & Levine, 1994*; *Kweon, Slade & Levine, 2011*). The starch granules of soft wheat mill easier than those of hard wheat resulting in intact granules that absorb less water (*Igrejas et al., 2002*). Good soft wheat produces high break flour yields with fine particle with minimal

Corresponding author
Gioconda Garcia-Santamaria,
gmgarcia@uce.edu.ec

damaged starch, and low arabinoxylan content so that the flour absorbs less water. The reduced water absorption capacity of soft wheat flour contributes to its functionality (*Finney & Bains, 1999*; *Souza, Graybosch & Guttieri, 2002*; *Kweon, Slade & Levine, 2011*). To fully characterize flour quality, it is important to evaluate flour protein (FP) and gluten functionality determined by specific combinations of high molecular weight subunits of glutenins associated with gluten strength (*Igrejas et al., 2002*).

Evaluation of soft wheat flour functionality is done by prediction tests. By combining alkaline water retention capacity (AWRC) and four solvent retention capacity (SRC) measurements, it is possible to determine the water absorption capacity of the flour as well as individual functional components that underlie it and determine flour functionality (*Slade & Levine, 1994*; *Gaines, 2000*; *Kweon, Slade & Levine, 2011*). Specifically, by the sodium carbonate (SO) SRC assesses the effect of damaged starch, the sucrose (SU) SRC assesses the effect of arabinoxylans, the lactic acid (LA) assesses the effect of glutenin characteristics, and the water (WA) assesses the overall water absorption capacity (*Slade & Levine, 1994*), making it easier to identify superior lines (*Souza, Graybosch & Guttieri, 2002*). The LA is a particularly useful measure as it assesses gluten strength and can be adjusted for the quantity of protein (adjusted LA, or ADLA) so that it relates to protein quality (*Gaines, 2000*). Soft wheat with high LA values have strong gluten and are suited for crackers and flat bread, while those with low LA have weaker gluten and are best suited for pastries (*Guttieri et al., 2001*). The LA has particular relevance to soft wheat as identity-preserved programs for strong-gluten soft red winter wheat exist in the eastern US (*Kweon, Slade & Levine, 2011*).

In addition to flour functionality, milling traits are an important component of soft wheat quality. Flour yield (FY) is a measure of straight grade flour from commercial mills with FY >72% being preferred. Softness equivalent (SE) and test weight (TW) are also considered is assessing soft wheat quality (*Finney & Andrews, 1986*; *Marshall et al., 1986*; *Finney & Bains, 1999*).

Allelic variation at loci encoding high molecular weight and low molecular weight glutenin subunits has a major influence on gluten strength (*Payne, Holt & Law, 1981*) (*Gupta, Singh & Shepherd, 1989*; *Rpusset et al., 1992*; *Nieto-Taladriz, Perretant & Rousset, 1994*; *Graybosch et al., 1996*). Glutenin subunits *GluD*x5 + *GluD*y10 confer strong dough mixing characteristics and good bread-making quality, while *GluD*x2 + *GluD*y12 are associated with weak dough and poor bread-making quality (*Payne, Holt & Law, 1981*; *Hamer, Weegels & Marseille, 1992*; *Manley, Randall & McGill, 1992*). Genes encoding glutenin subunits have been mapped to the short and long arms of homoeologous chromosomes 1A, 1B, and 1D (*Harberdt, Bartels & Thompsom, 1986*) and allele-specific primers can be used as markers to differentiate these alleles (*D'Ovidio & Anderson, 1994*; *Gale et al., 2003*). Loci associated with water absorption capacity have been identified in hard wheat (*Mansur et al., 1990*). Similarly, loci influencing FP, kernel hardness, and TW have been mapped (*Mattern et al., 1973*; *Blanco et al., 1996*; *Sourdille et al., 1996*; *Prasad et al., 1999*; *Perretant et al., 2000*; *Galande et al., 2001*; *Prasad et al., 2003*; *Turner et al., 2004*).

Previous correlation studies of soft wheat quality traits have already shown that flour damaged starch and arabinoxylan levels may be controlled by common genetic factors
(*Guttieri & Souza, 2003*; *Smith et al., 2011*; *Cabrera et al., 2015*; *Hoffstetter, Cabrera & Sneller, 2016*). Earlier studies about the heritability of SRCs in soft wheat have shown high heritability (*Guttieri & Souza, 2003*; *Smith et al., 2011*; *Cabrera et al., 2015*; *Hoffstetter, Cabrera & Sneller, 2016*). Common QTLs for AWRC and damaged starch were observed in a hard x soft population on chromosome 4DL (*Campbel et al., 2001*). *Smith et al. (2011)* and *Cabrera et al. (2015)* reported large effect QTL in SRWW for milling and baking quality associated with translocations on chromosomes 1B and 2B and that these effects were repeatable over populations. *Cabrera et al. (2015)* also presented evidence that QTL located on 1B and 2B affected SRWW quality even in the absence of the translocations. *Hoffstetter, Cabrera & Sneller (2016)* conducted an association analysis in SRWW and reported nine QTL for SRWW quality traits though the $r^2$ values were small (0.018 to 0.036).

SRC prediction tests are an efficient tool for predicting flour functionality. Knowledge about the underlying genetic control of these specific traits is necessary to supplement phenotypic selection. Identifying areas of the soft wheat genome harboring QTLs for functional end-use quality will assist in breeding and in understanding the genetic components of this suite of traits.

The main objective of this study was to identify QTLs related to quality traits in SRWW, and to broaden our knowledge of the underlying genetics of quality end-use traits.

## MATERIALS AND METHODS

### Plant materials

We used a recombinant inbred line (RIL) population consisting of 150 $F_4$-derived lines generated through single-seed descent from a cross of soft winter wheat lines Pioneer 26R46 by SS550. Parents and $F_{4:5}$ were grown in one replicate in an augmented block design in 2002, $F_{4:6}$ in two replicates during 2003, and $F_{4:7}$ in an augmented design in 2004 at the Ohio Agricultural Research and Developing Center (OARDC) in Wooster, OH, USA. Standard fertilizer applications were used with 150 kg ha$^{-1}$ of 18-46-0 (N-P-K) applied before planting and 65 kg ha$^{-1}$ of N applied in the spring. The plot size was a single 3 m row with 0.3 m space between rows. Replicates in 2003 were considered as environments (2003A and 2003B).

Parents were chosen based on the quality data report of the Soft Wheat Quality Laboratory (SWQL) of The United States Department of Agricultural Research Service (USDA, ARS) at Wooster, OH. At the time, Pioneer 26R46 was the highest quality ranking soft wheat cultivar due to its low water absorption capacity, high FY, large cookie diameter, high gluten strength, and *GluD*x5 + *GluD*y10 alleles. The parent SS550 (VA96W-247) has moderate quality, low FP, very soft texture, moderate gluten strength, and high AWRC.

### Quality determination

Quality analysis was conducted in the USDA SWQL of Wooster, OH, USA on the single replication in 2002, separate grain samples from each replication in 2003 (A y B) and a single grain sample pooled from both reps in 2004. The reason for this sampling is that in 2002 there was only enough grain for one plot; in 2003 we planted two plots though
the analysis indicated there was little error within a field. Thus in 2004 grain from the two reps was pooled to form one sample that was assayed. Grain from the parents and RILs was threshed, cleaned, tempered to 14% moisture, and milled in a Quadrumat junior mill (American Association of Cereal Chemists (AACC) method 26–50) to determine milling and flour quality characteristics. Milling traits (FY, TW, and SE), FP, and AWRC were measured using standard procedures as described by AACC methods 39–11 and 56–10 (*AACC, 1983*). The SRCs were measured according to AACC method 56–11 (*Gaines, 2000*).

## Statistical analysis of phenotypic traits

Variation in the parents was determined using analysis of variance (ANOVA). Phenotypic data from parents and RILs for ten quality traits from four environments (2002, 2003A, 2003B, and 2004) was analyzed using Proc GLM of Statistical Analysis System (SAS) v.9.1 (SAS Institute, Cary, NC, USA) using the model:

$$y_{ij} = u + g_i + e_j + \text{error}_{ij}$$

where $g_i$ is the effect of the ith genotype and $e_j$ is the effect of the jth environment. Genotype and environment effects were considered random. This analysis was used to estimate an LSD ($P < 0.05$) to test whether RILs differed from their parents as well as other comparisons. We estimate variance components with PROC MIXED (SAS v9.1) (SAS Institute, Cary, NC, USA) using just RIL data to test the significance of RIL effects. Best linear unbiased predictors were obtained for each RIL and were used in mapping and in for correlation analysis for the ten quality parameters. The RIL effects were highly correlated between environments so data were combined over environments for analysis. Entry-mean heritability (H) was calculated using only RIL data as:

$$H = \sigma_g^2 / \sigma_{g+}^2 (\sigma_{\text{error}}^2 / 4)$$

where $\sigma_g^2$ and $\sigma_{\text{error}}^2$ are the genetic and error variance, respectively.

## Linkage map

Parents were screened for polymorphism using 700 Single Sequence Repeats (SSR) primers previously published (*Röder et al., 1998*; *Cregan, Ward & Gill, 2001*; *Gupta et al., 2002*). The population was genotyped with 107 SSR markers that were polymorphic between the parents. The *GluD*x5 allele-specific primer developed by *Gale et al. (2003)* was also included.

Genetic linkage maps were constructed with JoinMap 3.0 (*Van Ooijen & Voorrips, 2001*). Grouping of similar loci was based upon the test for independence and was done at several significance levels of the logarithm of the odds (LOD) scores. Linkage groups were constructed at a probability of 0.0001 followed by the 'ripple' command to refine the order of markers and place the marker loci in a linkage group.

## QTL analysis

QTL positions in the genome were calculated using MapQTL 4.0 (*Van Ooijen, 2002*) with composite interval mapping with the maximum likelihood approach. The components (Q) of a mixture depending on the QTL genotype, which would be Q = 3 in the case of the

**Table 1** Parental and population means, and maximum and minimum values for each quality trait of 150 soft red winter wheat RIL combined over four environments.

| Trait | RIL mean | Pioneer 26R46 | SS550 | RIL maximum | RIL minimum |
|---|---|---|---|---|---|
| TW (kg m$^{-3}$) | 778 | 767[*] | 789 | 833 | 733 |
| AWRC (%) | 534 | 52[*] | 57 | 61 | 49 |
| FP (g kg$^{-1}$) | 103 | 99[ns] | 102 | 120 | 83 |
| LA (g kg$^{-1}$) | 949 | 1,003[*] | 922 | 1,346 | 650 |
| ADLA (g kg$^{-1}$) | 860 | 943[*] | 838 | 1,250 | 567 |
| WA (g kg$^{-1}$) | 514 | 491[*] | 539 | 559 | 475 |
| SU (g kg$^{-1}$) | 833 | 815[*] | 895 | 963 | 752 |
| SO (g kg$^{-1}$) | 626 | 597[*] | 674 | 717 | 562 |
| FY (%) | 72 | 72[*] | 68 | 74 | 63 |
| SE (%) | 53 | 55[*] | 56 | 61 | 39 |

**Notes.**

[*]Indicates a significant difference between parental means at the $P < 0.05$ level.

[ns]not significant

TW, test weight; AWRC, alkaline water retention capacity; FP, flour protein; LA, lactic acid SRC; ADLA, adjusted LA; WA, water SRC; SU, sucrose SRC; SO, sodium carbonate SRC; FY, flour yield; SE, softness equivalent.

RIL. The component distributions are assumed to be normal, and the Haldane mapping function was used, which assumes that recombination events are mutually independent. QTLs are calculated under the alternative hypothesis that a single QTL is segregating. The likelihood (LOD) is calculated at each iteration, and QTLs were considered to be those regions having LOD $\geq$ 2.8. The functional tolerance value and the maximum number of interactions used were 200.

# RESULTS

## Quality testing

The two parents differed significantly nine traits but not for FP (Table 1). The RILs exhibited a continuous distribution and transgressive segregants were observed for all traits (Figs. 1–4). Minimum and maximum means of RILs exceeded the means of the two parental lines, indicating new allelic combinations for all traits (Table 1). Significant phenotypic variation existed among RILs for all quality parameters. Variation between environments was significant for all traits except for FP (Table 2). Variance component analysis showed that genotypic variance was higher than environmental variance for all traits except TW and AWRC. Heritability of the ten quality traits ranged from 0.67 to 0.90 (Table 3).

The RILs means across environments were used for correlation analysis. Significant positive correlations among RIL means for quality traits ranged from 0.17 to 0.88, and significant negative correlations ranged from −0.10 to −0.76 (Table 4). The WA, SO, SU and WARC were highly positively correlated to one another and all were highly negatively correlated to FY.

## Linkage map and QTL analysis

The 107 markers were assigned to 18 linkage groups (Fig. 5). The positions and order of the markers were verified and in agreement with earlier published maps (*Röder et al., 1998*;

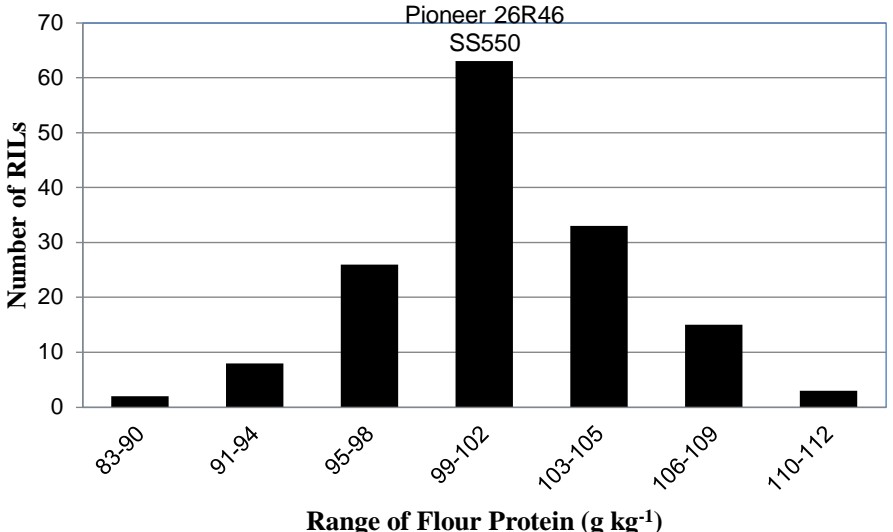

**Figure 1** Distribution of average flour protein values for wheat recombinant inbred lines and their parents.

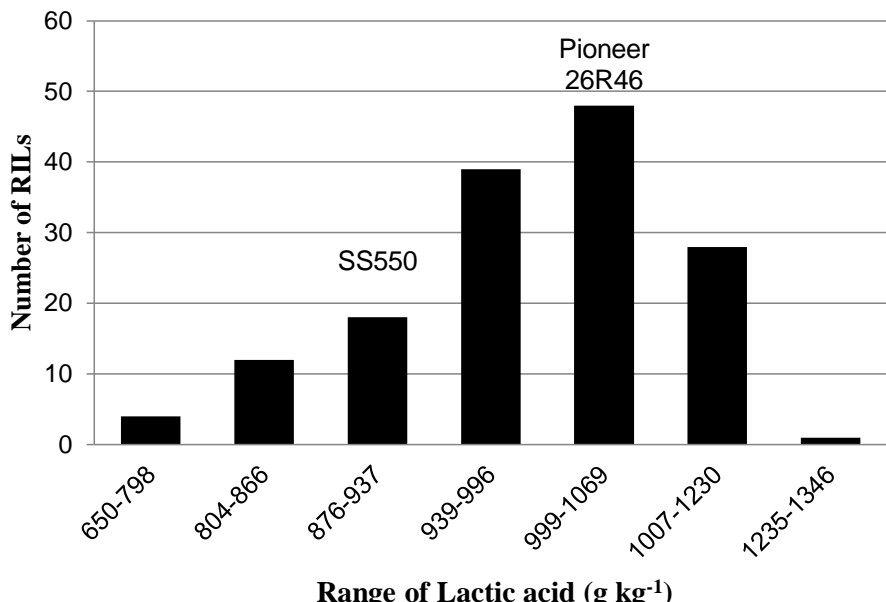

**Figure 2** Distribution of average lactic acid SRC values for wheat recombinant inbred lines and their parents.

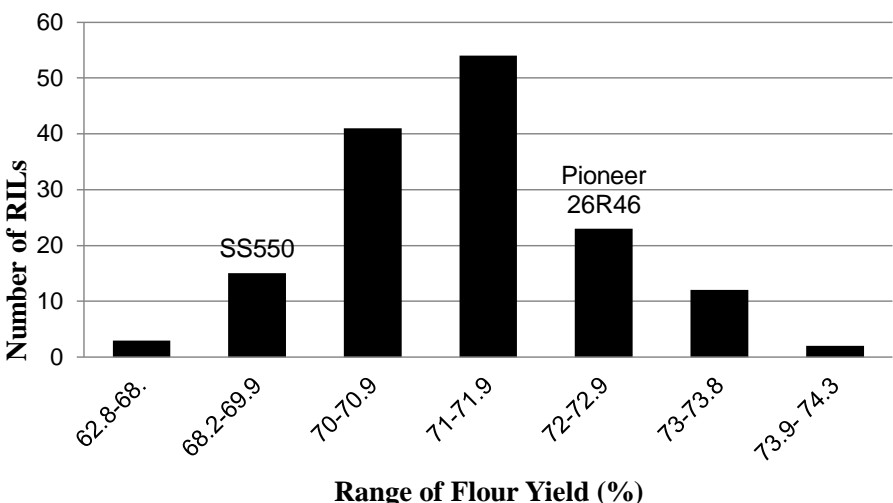

**Figure 3** Distribution of average flour yield values for wheat recombinant inbred lines and their parents.

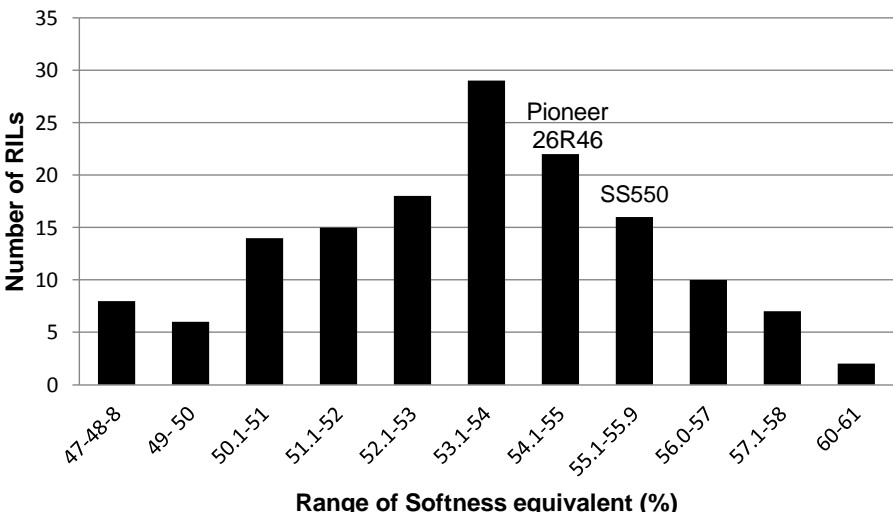

**Figure 4** Distribution of average softness equivalent values for wheat recombinant inbred lines and their parents.

*Gupta et al., 2002*). Eight markers deviated significantly from the expected segregation ratio. The dominant marker for HMW-glutenin subunit *GluDx*5 on chromosome 1D showed segregation distortion.

Eight chromosome regions showed QTLs associated with one or more of the 10 quality traits (Table 5). In total there were 28 significant trait-marker associations. One region of chromosome 1A affected six of 10 traits including traits for water absorption capacity (AWRC, SO), gluten strength (LA, ADLA), and milling quality (FY, SE). This region had the greatest effect of all regions for AWRC, LA, ADLA, and SO. One region of chromosome

**Table 2 Sum of squares of the combined ANOVA for ten quality parameters of 150 soft red winter wheat RIL from four environments.**

| Solvent retention capacities | Source of variation | |
|---|---|---|
| | Environment | RIL |
| TW (kg m$^{-3}$) | 630$^{**}$ | 4.9$^{**}$ |
| AWRC (%) | 468$^{**}$ | 6.8$^{**}$ |
| FP (g kg$^{-1}$) | 3$^{ns}$ | 5.8$^{**}$ |
| LA (g kg$^{-1}$) | 46$^{**}$ | 15.2$^{*}$ |
| ADLA (g kg$^{-1}$) | 60$^{**}$ | 15.4$^{**}$ |
| WA (g kg$^{-1}$) | 24$^{**}$ | 8.8$^{**}$ |
| SU (g kg$^{-1}$) | 162$^{**}$ | 6.5$^{**}$ |
| SO (g kg$^{-1}$) | 73$^{**}$ | 9.1$^{**}$ |
| FY (%) | 180$^{**}$ | 6.4$^{**}$ |
| SE (%) | 394$^{**}$ | 8.5$^{**}$ |

Notes.
$^{*}$Indicates significance at $P < 0.05$.
$^{**}$Indicate significance at $P < 0.001$.
$^{ns}$not significant.

TW, test weight; AWRC, alkaline water retention capacity; FP, flour protein; LA, lactic acid SRC; ADLA, adjusted LA; WA, water SRC; SU, sucrose SRC; SO, sodium carbonate SRC; FY, flour yield; SE, softness equivalent.

**Table 3 Heritability and variance components across environments for ten quality parameter in soft red winter wheat.**

| Solvent retention capacities | Variance components | | | | |
|---|---|---|---|---|---|
| | $\sigma^2_{env}$ | $\sigma^2_g$ | $\sigma^2_{error}$ | $\sigma^2_g/\sigma^2_{error}$ | $h^2$ |
| TW (kg m$^{-3}$) | 1.90 | 0.46 | 0.40 | 1.10 | 0.81 |
| AWRC (%) | 2.40 | 1.00 | 2.10 | 2.80 | 0.67 |
| FP (g kg$^{-1}$) | 0.0001 | 0.20 | 0.20 | 1.00 | 0.80 |
| LA (g kg$^{-1}$) | 162 | 830 | 341 | 0.50 | 0.91 |
| ADLA (g kg$^{-1}$) | 157 | 842 | 298 | 2.40 | 0.92 |
| WA (g kg$^{-1}$) | 0.05 | 1.20 | 0.80 | 1.50 | 0.90 |
| SU (g kg$^{-1}$) | 2.50 | 6.70 | 4.80 | 1.40 | 0.85 |
| SO (g kg$^{-1}$) | 1.70 | 4.20 | 2.30 | 1.90 | 0.90 |
| FY (%) | 0.30 | 1.10 | 0.90 | 1.30 | 0.84 |
| SE (%) | 2.70 | 4.90 | 2.20 | 2.20 | 0.90 |

Notes.
TW, test weight; AWRC, alkaline water retention capacity; FP, flour protein; LA, lactic acid SRC; ADLA, adjusted LA; WA, water SRC; SU, sucrose SRC; SO, sodium carbonate SRC; FY, flour yield; SE, softness equivalent.

1B affected five traits with a large effect on LA and ADLA ($r^2 = 0.33$–0.34). Regions of chromosomes 2B, 3B, 4A, and 6B all accounted for greater than 14.9 % of the phenotypic variation for at least one trait.

For all 28 trait-marker associations the allele that increased the trait came from the parent with the higher phenotypic value (Tables 1 and 5). There were four QTL for water absorption traits (AWRC, WA, SU, SO) on four chromosomes (1A, 1B, 4A, and 6B) (Table 5, Fig. 5). Three of these four regions affected more than one water absorption capacity trait. In all 10 trait-marker combinations for these traits the alleles from Pioneer

**Table 4  Pearson's correlation coefficients for ten quality parameters of 150 soft red winter wheat RIL.**

|      | FP | LA | ADLA | AWRC | WA | SU | SO | FY | SE |
|------|----|----|------|------|----|----|----|----|----|
| TW   | 0.17* | 0.18* | ns | ns | 0.24*** | ns | 0.20* | ns | ns |
| FP   |    | 0.36** | ns | ns | 0.23** | ns | ns | ns | −0.26*** |
| LA   |    |    | 0.95*** | −0.22** | ns | ns | ns | ns | 0.05** |
| ADLA |    |    |      | −0.10** | ns | 0.33** | ns | ns | ns |
| AWRC |    |    |      |      | 0.71*** | 0.66*** | 0.88*** | −0.64*** | 0.43*** |
| WA   |    |    |      |      |    | 0.80*** | 0.79*** | −0.63*** | ns |
| SU   |    |    |      |      |    |    | 0.79*** | −0.75*** | ns |
| SO   |    |    |      |      |    |    |    | −0.76*** | 0.50*** |
| FY   |    |    |      |      |    |    |    |    | −0.50*** |

**Notes.**
*$P < 0.05$.
**$P < 0.01$.
***$P < 0.001$.
ns, not significant.
TW, test weight; AWRC, alkaline water retention capacity; FP, flour protein; LA, lactic acid SRC; ADLA, adjusted LA; WA, water SRC; SU, sucrose SRC; SO, sodium carbonate SRC; FY, flour yield; SE, softness equivalent.

26R46 decreased the trait value and would be the desired allele. QTL for milling traits (FY, SE) were detected on chromosomes 1A, 1B, 2B, 3B, 4A, and 6B. For each the desired allele for SE came from SS550 while the desired alleles from FY came from Pioneer 26R46. There were two regions (1A and 1B) associated the LA and ADLA and neither were associated with FP. QTL for FP were detected on three regions (2B, 5A, and 5D). Two regions were associated with TW on chromosomes 1B and 6B with the desired allele coming from SS550.

## DISCUSSION

The mapping population derived from a cross of two elite SRWW lines offered the opportunity to study the genetic determination and the identification of important areas of the genome containing QTLs associated with specific components related to flour milling and functional quality. In our study, the parents differed significantly for nine of 10 traits (Table 1) and their phenotypes were in general correspondence to the values in 2005 report of the Soft Wheat Quality Laboratory (SWQL) of The United States Department of Agricultural Research Service (USDA, ARS) at Wooster, OH. The RILs showed a continuous phenotypic variation and transgressive segregation. Heritability for all traits ranged from 0.67 to 0.90 (Table 3). Others have reported similar heritability values for soft wheat quality traits (*Baenziger et al., 1985*; *Basset, Allan & Rubenthaller, 1989*; *Cabrera et al., 2015*; *Guttieri & Souza, 2003*; *Hoffstetter, Cabrera & Sneller, 2016*; *Smith et al., 2011*; *Souza, Graybosch & Guttieri, 2002*).

A total of 28 marker-trait associations were detected from eight chromosome regions (Table 5). Some regions of the genome contained coincident QTLs associated with more than one trait. The coincident QTL often corresponded to trait correlations (Table 4). The water absorption capacity traits AWRC, WA, SU, and SO were all positively correlated as has been reported by others for soft wheat (*Guttieri & Souza, 2003*; *Ram et al., 2005*; *Smith et al., 2011*; *Cabrera et al., 2015*; *Hoffstetter, Cabrera & Sneller, 2016*). There were 10

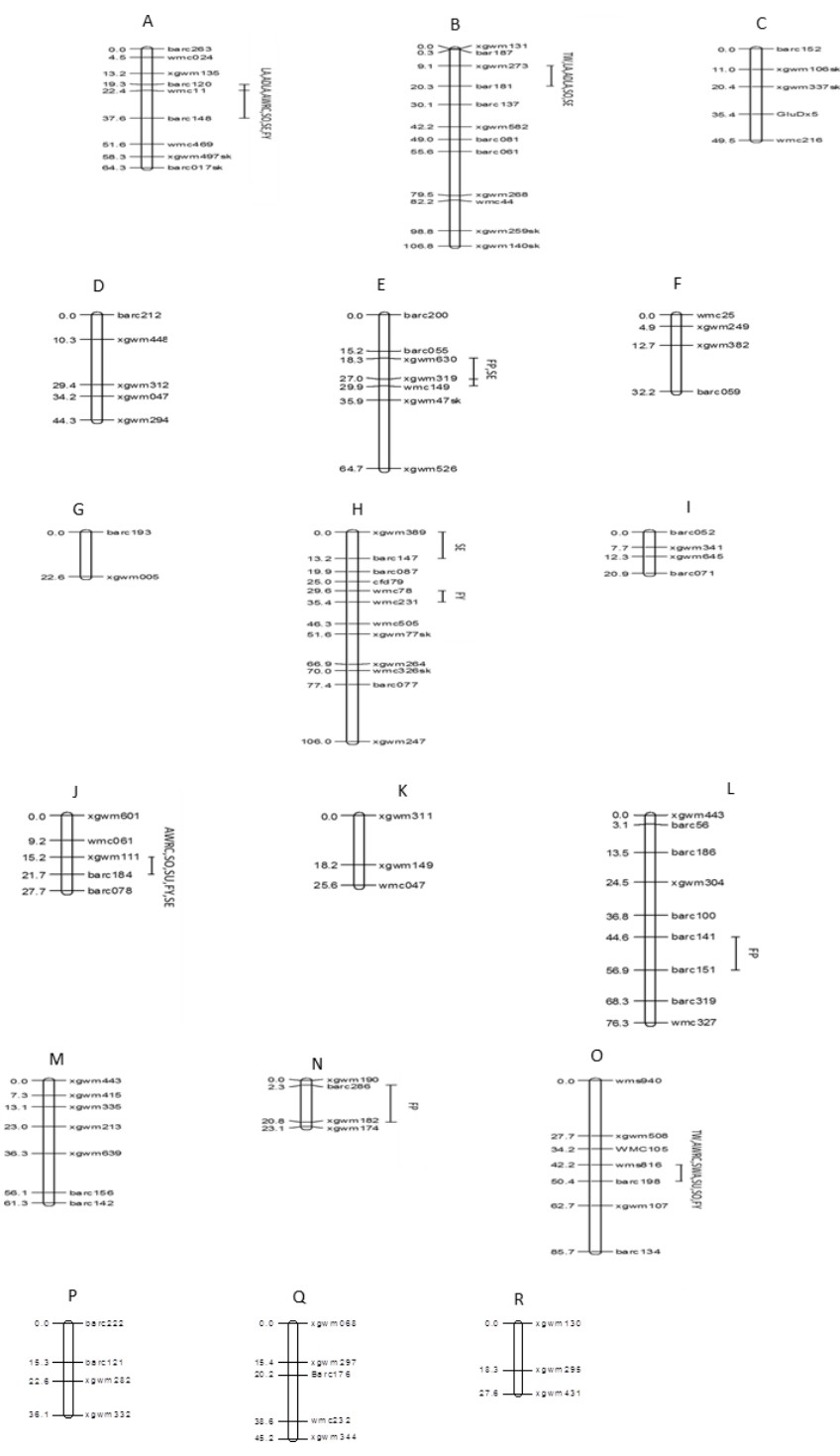

**Figure 5** **Genomic locations of QTLs for ten quality traits in a mapping population of 150 soft red winter wheat recombinant inbred lines.** (A) 1A, (B) 1B, (C) 1D, (D) 2A, (E) 2B, (F) 2D, (G) 3A, (H) 3B, (I) 3D, (J) 4A, (K) 4B, (L) 5A, (M) 5B, (N) 5D, (O) 6B, (P) 7A, (Q) 7B, (R) 7D. Map distances (cM) are show on the left and marker names are show on the right. Marker positions were deduced by comparison with other maps. Segregation distortion is indicated with (sk). TW, test weight; AWRC, alkaline water retention capacity; FP, flour protein; LA, lactic acid SRC; ADLA, adjusted LA; WA, water SRC; SU, sucrose SRC; SO, sodium carbonate SRC; FY, flour yield; SE, softness equivalent.

**Table 5** Chromosomes with QTLs controlling quality traits detected by composite interval mapping in soft red winter wheat.

| Chromosome | Interval | Trait | % variation | LOD | Additive effect of Pioneer 26R46 |
|---|---|---|---|---|---|
| 1AL | Barc120–Barc148 | LA | 42.6 | 9.0 | 6.5 |
| | Barc120–Barc148 | ADLA | 36.0 | 8 .0 | 6.0 |
| | Barc120–Barc148 | AWRC | 25.0 | 6.0 | −0.6 |
| | Barc120–Barc148 | SO | 26.0 | 6.0 | −1.3 |
| | Barc120–Barc148 | SE | 25.0 | 5.0 | −1.2 |
| | Barc120–Barc148 | FY | 15.0 | 3.0 | 0.4 |
| 1BL | Barc181-Barc137 | TW | 17.0 | 5.0 | −0.5 |
| | Xgwm273-Barc137 | LA | 33.0 | 9.0 | 5.6 |
| | Xgwm273-Barc137 | ADLA | 34.0 | 9.0 | 5.2 |
| | Xgwm273-Barc137 | SO | 11.0 | 2.8 | −0.6 |
| | Barc181-Barc137 | SE | 12.0 | 3.4 | −0.7 |
| 2B | Xgwm630-Wmc149 | FP | 16.0 | 5.0 | −0.2 |
| | Xgwm630-Wmc149 | SE | 7.5 | 2.3 | 0.6 |
| 3B | Barc147–Cfd79 | SE | 20.0 | 5.0 | −1.0 |
| | Wmc78-Wmc231 | FY | 10.0 | 3.2 | 0.4 |
| 4A | Xgwm111-Barc184 | AWRC | 10.0 | 2.9 | −0.4 |
| | Xgwm111-Barc184 | SU | 12.0 | 3.7 | −0.9 |
| | Xgwm111-Barc184 | SO | 15.0 | 4.6 | −0.8 |
| | Xgwm111-Barc184 | SE | 8.4 | 2.8 | −0.7 |
| | Xgwm111-Barc184 | FY | 13.5 | 4.2 | 0.4 |
| 5A | Barc141-Barc151 | FP | 10.0 | 2.9 | −0.17 |
| 5D | Barc286-Xgwm182 | FP | 12.7 | 2.5 | −0.2 |
| 6B | Barc198-Wms816 | TW | 12.5 | 3.8 | −0.3 |
| | Barc198-Wms816 | AWRC | 12.0 | 4.0 | −0.5 |
| | Barc198-Wms816 | WA | 22.0 | 7.8 | −0.6 |
| | Barc198-Wms816 | SU | 31.0 | 7.8 | −1.6 |
| | Barc198-Wms816 | SO | 18.0 | 6.0 | −0.9 |
| | Barc198-Wms816 | FY | 10.0 | 3.0 | 0.4 |

**Notes.**
TW, test weight; AWRC, alkaline water retention capacity; FP, flour protein; LA, lactic acid SRC; ADLA, adjusted LA; WA, water SRC; SU, sucrose SRC; SO, sodium carbonate SRC; FY, flour yield; SE, softness equivalent.

marker-trait associations from four regions for these traits and only one was not coincident with another. In all cases, the allele from Pioneer 26R46 was the desired allele as it decreased water absorption, as would be predicted by the parental phenotypes for these traits (Table 1). The results suggest that these markers along with the parental phenotype could be used as good predictors of end-use functionality. FY was negatively correlated with the water absorption capacity traits as has been reported by others (*Smith et al., 2011*; *Cabrera et al., 2015*; *Hoffstetter, Cabrera & Sneller, 2016*). Three of the four regions associated with water absorption traits were also associated with FY. As expected from the correlations, if a QTL allele decreased water absorption it increased FY. This has been reported by others is SRWW (*Smith et al., 2011*; *Cabrera et al., 2015*). Earlier studies explained that soft wheat

genotypes with less damaged starch and lower arabinoxylan content have higher flour extraction (*Guttieri et al., 2001*). *Finney & Bains (1999)* explained that low FY cultivars that perform very poorly during milling, have increased levels of damaged starch, and consequently would have increased water absorption.

Loci associated with FP mapped on chromosomes 2B, 5B and 5D. QTLs on chromosome 2B co-segregated for SE: SE was negatively correlated (−0.26) with FP. A negative correlation of −0.45 between these traits was also observed in hard wheat (*Gross, Bervas & Charmet, 2004*) and others have reported a negative correlation between these traits in soft wheat (*Smith et al., 2011*; *Cabrera et al., 2015*; *Hoffstetter, Cabrera & Sneller, 2016*). Genetic studies of kernel hardness in bread wheat indicated that phenotypic expression of kernel hardness was tightly linked with FP (*Galande et al., 2001*), but additional related traits such as arabinoxylan content also played an important role in kernel hardness (*Bettge & Morris, 2000*).

A positive significant correlation between FP and LA (0.36) was detected though the QTL for these two traits were not coincident in this study. Positive correlations between these two traits have been previously reported (*Guttieri et al., 2001*; *Knott, Van Sanford & Souza, 2009*). However, in a study of three soft wheat populations, just one population showed a positive correlation (0.47) between these traits (*Guttieri & Souza, 2003*). Lack of association of these two traits was also observed in a study of soft white wheat (*Guttieri et al., 2001*). This association of LA and FP has been explained by the effect of high molecular weight (HMW) glutenin subunits, which are part of the total FP. A study of the influence of storage protein alleles on quality traits determined that HMW glutenin alleles encoded at *Glu-A1* and *Glu-B1* cause significant differences in quality parameters related to gluten strength (extensibility and strength), flour yield, and FP while HMW glutenin subunits *GluD*x2 + *GluD*y12, *GluD*x3 + *GluD*y12, and *GluD*x4 + *GluD*y12, *GluD*x2 + *GluD*y12 at the *Glu-D1* locus, have no effect on extensibility, strength, flour yield, FP, and mixograph parameters (*Igrejas et al., 2002*). In our study gluten strength functionality measured by LA was independent to the other SRC tests, yet these traits have been reported to be positively associated (*Guttieri et al., 2001*). The lack of correlation observed in our study was probably because LA is a specific test for the glutenin network swelling behavior (*Kweon, Slade & Levine, 2011*). Major QTLs with the largest effect on LA and consequently gluten strength were on chromosomes 1A with LOD 9 that explained 42.6% of LA variation and QTLs on chromosome 1B with LOD 9 that explained 33% of the variation in LA. Loci on chromosomes 1A and 1B were also important contributors of additive effects with an increase of 6.5 and 5.6 percent, respectively. The two regions affecting LA and ADLA on chromosomes 1A and 1B in our study are likely co-located with the *Glu-A1* and *Glu-B1* loci.

QTL on chromosome 2B were previously found in bread wheat recombinant substitution lines and in a soft × hard wheat population (*Campbel et al., 2001*; *Turner et al., 2004*). *Cabrera et al. (2015)* and *Smith et al. (2011)* reported that 2B was one of the key chromosomes controlling soft wheat quality, along with 1B. Pioneer 26R46 carries the 1BL:1RS translocation that has been shown to have a large effect on soft wheat quality (*McKendry et al., 1996*; *McKendry, Tague & Ross, 2001*; *Cabrera et al., 2015*). The IBL:1RS is often detrimental to quality though Pioneer 26R46 shows superior quality anyways. The

effect of 2B on quality traits can be partly attributed in some crosses to the *T. timopheevi* translocation associated with *Sr36* (*Allard & Shands, 1954*; (*Tsilo, Jin & Anderson, 2008*)) and to allelic variation for sucrose synthase (*Cabrera et al., 2015*).

## CONCLUSIONS

This study validates some previous findings in soft wheat that chromosomes 1B and 2B are important to soft wheat quality. Previous studies have not shown the regions of 1A to be as important for soft wheat quality as we are reporting here. Perhaps some novel alleles from Pioneer 26R46 are causing the large effects associated with 1A and contributing to the very high quality of grain from Pioneer 26R46.

Many applications of MAS begin with mapping a desired allele from a donor with a particular marker allele: MAS is executed in a lineage derived from the donor, thus assuring that identity-by-state at the marker reflects identity by descent at the QTL. In this study the parents with the favorable phenotype always contributed the favorable alleles at key QTL, similar to the results from *Cabrera et al. (2015)*. This suggests that in other crosses that one could select for superior progeny by selecting for the best parent's marker alleles at the key loci (say 1A, 1B, 2B, 6B), even without prior determination of the effects of these marker alleles from these parents. Our findings support the similar conclusion made by *Cabrera et al. (2015)*. Thus, instead of using marker-assisted selection to bred for a QTL derived from a single ancestor, one could possibly use MAS in any cross by selecting for markers from the superior parent, regardless of their ancestral source.

### Funding
The authors received no funding for this work.

### Competing Interests
The authors declare there are no competing interests.

### Author Contributions
- Gioconda Garcia-Santamaria conceived and designed the experiments, performed the experiments, analyzed the data, contributed reagents/materials/analysis tools, prepared figures and/or tables, authored or reviewed drafts of the paper, approved the final draft.
- Duc Hua performed the experiments, contributed reagents/materials/analysis tools, authored or reviewed drafts of the paper, approved the final draft.
- Clay Sneller conceived and designed the experiments, analyzed the data, prepared figures and/or tables, authored or reviewed drafts of the paper, approved the final draft.

### Data Availability
The raw data are provided in Supplemental Information 1.

## Supplemental Information

Supplemental information for this article can be found online at http://dx.doi.org/10.7717/peerj.4498#supplemental-information.

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
