# Peer review of "Quantitative trait loci associated with soft wheat quality in a cross of good by moderate quality parents"

_PeerJ, doi:10.7717/peerj.4498_

## Round 0.1 · original submission · Major Revisions

The major concern about this manuscript is the small population size and low marker density. Nevertheless it could be considered for publication if the suggestions of both reviewers are implemented in an improved manuscript version. Please, also include new literature.

Reviewer 1 ·

Basic reporting

• the manuscript is written clearly and in good english

Experimental design

• The size of the population is quite small, unusual for QTL studies nowadays
• The choice of genetic markers and the number of markers used in this manuscript is not suitable for QTL studies nowadays
• The phenotypic data is old, from 2002 to 2004 and grown in very small plots (only one row)
• In general, there is no reason not to use old phenotypic data but you would rather combine it with new genetic data or at least a current statistical method, but this is not the case in this manuscript
M&M part has no novelty impact to the current scientific questions in QTL studies

Validity of the findings

• The low resolution of the map makes QTL location impossible, traits can only be associated to a chromosome, and this is not the state of the art anymore.
• A search for QTL quality in soft wheat in google scholar results in newer literature with greater populations and higher amount of genetic markers and therefore a better genetic resolution
• Conclusion within the discussion are missing
• “The take home massage is that you would just need to identify the better parent of a cross and select for the allele of the better parent”
o this is not a novel assumption

Additional comments

Introduction: General comments
• Explain abbreviations within the introduction
• Be consistent with the abbreviations, is it LA or is it LA SRC (65-67)
• Your introduction is quite detailed concerning the traits under investigation but you do not write anything about QTL mapping in your introduction
• Please include a section about QTL mapping, population size, model selection, choice of parents for crosses in your introduction


Material and Methods: General comments
• M&M in general is quite short
• 115: it is not evident, if you had 2 replications in 2004 (in 127 you say you pooled the grain samples of both reps in 2004), please be more precise
• 121: is Pioneer 26R46 still the cultivar with the highest quality in soft winter wheat? Be more precise and critical with the choice of parents at that time when the population was constructed, especially, because your population you are using is quite old.
• 137: Phenotypic means? To my knowledge, this is not a statistical term, please explain how you calculated means. Why did you not use LSmeans calculated in SAS, this is statistically verified
• 140: please check punctuation
• 143: and environment
• 146: formula does not fit to the results, include environment
• 159: please check punctuation
• 165 please check grammar

Results: General comments
• 170: either include figure in supplementary data, or delete the sentence, if you do not show the data then you cannot prove your statement
• 182-184: part of discussion

Discussion: General comments
• 222: eight chromosomes
• 241: Language
• 271: which trait?
• Further Conclusion within the discussion are missing



Table 1, Table 2:
• Please sort traits in alphabetic order
• Please have the same number of digits after the value column wise

·

Basic reporting

1. The manuscript is well written and readable. However please correct the subject -verb or subject article agreements as in line 74; 76, 104; 151, 169,
2. Table 1: should include the best RILs
3. Include the association of quality traits with agronomic traits in the result section
4. In Table 4, I could not see the columns for the years though it is titled across years. It should be average over the years.
5. List the best transgressive RILs in terms of quality as compared to the best check.
6. Show the frequency distribution in graph of the RILS in relation to the two parents for the important traits such as FY, FP, LA.

Experimental design

characterize the environments ( soil type, PH, Temperature, rainfall) in a Table as these factors affect quality traits in wheat. .

Validity of the findings

The experiment confirms the importance of chromosome 1B and 2B for soft wheat quality. It also reports the identification of new QTLs on chromosome 1A.

However, this should be validated further.

Additional comments

1. The abstract is too long. Please summarize it and just indicate the important part of your methodology, key findings and recommendations.
2. In the discussion part, please review the list of previously reported genes and references in a Table.
3. Check again some of the missing words/articles in the sentences.

---

## Round 0.2 · accepted · Accept

The revised manuscript can be accepted for publication.

Reviewer 1 ·

Basic reporting

In general, the english is fine

please change: trait-marker associatin = marker-trait association

Experimental design

The research question, the population design and especially the choice of markers is not relevant to this time anymore

Validity of the findings

no comment

Additional comments

no comment

·

Basic reporting

I have no further comment

Experimental design

No further comment

Validity of the findings

I have no further comment